# Association of Genes *TRH*, *PRL* and *PRLR* with Milk Performance, Reproductive Traits and Heat Stress Response in Dairy Cattle

**DOI:** 10.3390/ijms26051963

**Published:** 2025-02-24

**Authors:** Qianhai Fang, Hailiang Zhang, Qing Gao, Lirong Hu, Fan Zhang, Qing Xu, Yachun Wang

**Affiliations:** 1Key Laboratory of Animal Genetics, Breeding and Reproduction, Ministry of Agriculture and Rural Affairs of China, National Engineering Laboratory for Animal Breeding, State Key Laboratory of Animal Biotech Breeding, College of Animal Science and Technology, China Agricultural University, Beijing 100193, China; 2Institute of Life Sciences and Bio-Engineering, Beijing Jiaotong University, Beijing 100044, China

**Keywords:** milk performance, fertility, heat stress, prolactin, prolactin receptor, thyrotropin-releasing hormone

## Abstract

In our previous study, we found that changes in plasma prolactin (PRL) concentration were significantly associated with heat stress in dairy cows, and that PRL plays an important role in milk performance. Microarray sequencing revealed that thyrotropin releasing hormone (*TRH*) and prolactin receptor (*PRLR*), two genes important for *PRL* expression or function, may affect milk performance, reproduction, and heat stress response in dairy cattle. In this study, we further validated the genetic effects of the three genes in Chinese Holsteins. The potential variants within the three genes were first detected in 70 Chinese Holstein bulls and then screened in 1152 Chinese Holstein cows using the KASP (Kompetitive allele-specific PCR) method. In total, 42 variants were identified. Further, 13 SNPs were retained for KASP genotyping, including 8 in *TRH*, 3 in *PRL*, and 2 in *PRLR*. Using SNP-based association analyses, the multiple significant (*p* < 0.05) associations of these 13 SNPs with milk performance, reproduction, and heat stress response traits were found in the Holstein population. Furthermore, linkage disequilibrium analysis found a haplotype block in each of the *TRH* and *PRL* genes. Haplotype-based association analyses showed that haplotype blocks were also significantly (*p* < 0.05) associated with milk performance, reproduction, and heat stress response traits. Collectively, our results identified the genetic associations of *TRH*, *PRL*, and *PRLR* with milk performance, reproduction, and heat stress response traits in dairy cows, and found the important roles of SNP g.55888602A/C and g.55885455A/G in *TRH* in all traits, providing important molecular markers for genetic selection of high-yielding dairy cows.

## 1. Introduction

In dairy breeding, milk performance and reproduction are the most important economic and functional traits. While milk performance has been greatly improved by intensive selection, it has been accompanied by a generally observed decline in the reproductive performance of dairy cows [1]. Moreover, with the rising global temperature, heat stress is causing extensive welfare and economic losses in the worldwide dairy cattle industry, especially affecting reproductive performance [2,3,4,5]. Due to the highly polygenic characteristics and unfavorable correlations, genetic selection for reproductive and heat tolerance traits is relatively slow without compromising milk performance. By integrating key variants in tight linkage disequilibrium (LD) with causal mutations into genomic prediction, genetic improvement in reproductive and heat tolerance traits can be accelerated through improved selection accuracy. Importantly, selection accuracy can be further enhanced by considering genomic relationships among individuals.

In the past few decades, quantitative trait locus (QTL) mapping, candidate gene analysis, and genome-wide association studies have been widely used to identify the potential causal genes or mutations related to economically important traits in dairy cattle [6,7]. The genetic gain of dairy cows achieved through molecular breeding for genomic selection has greatly surpassed that of conventional selection methods [8]. However, genomic selection relies on single-nucleotide polymorphism (SNP) information across the whole genome to estimate the effects of different chromosome segments and subsequently calculate individual genomic estimated breeding values (GEBVs) [9]. Therefore, identifying candidate genes and molecular markers associated with key traits is crucial. Despite this progress, there is still a need for more knowledge about molecular markers for economically important traits in dairy cows, and further studies are needed to identify key loci associated with these traits to enhance selection accuracy.

Prolactin (PRL) is a peptide hormone secreted by the cells of the anterior pituitary gland [10]. It plays a crucial role in regulating the initiation and maintenance of lactation, as well as the reproductive process [11,12]. In addition, PRL is recognized as a sensitive hormone in response to heat stress and is essential for maintaining environmental homeostasis in the body [13,14,15]. Our previous research found that changes in plasma PRL concentration were significantly associated with heat stress in dairy cows [16]. The bovine *PRL* gene is located on chromosome 23, with 45 SNPs within exons reported in the Ensemble database [17,18,19]. Previous studies have revealed strong effects of *PRL* on milk performance traits in Romanian cattle, Czech Fleckvieh cattle, and Italian Mediterranean river buffalo [11,20,21]. Further study is needed to clarify the associations between genetic polymorphisms in *PRL* and milk performance, reproductive traits, and heat stress. As critical regulatory components in prolactin signaling, thyrotropin-releasing hormone (TRH) and prolactin receptor (PRLR) also play essential roles in regulating milk performance, growth, and development [22,23]. However, the genetic effects of known SNPs in *PRLR* on milk performance, reproductive traits, and heat stress in dairy cows remain unclear, and polymorphisms of the *TRH* gene and their association with various traits have not yet been reported in dairy cattle.

Collectively, the objectives of this study were (1) to screen for polymorphisms of the *PRL*, *PRLR*, and *TRH* genes in a large Holstein population; (2) to validate the genetic effects of these three genes on milk performance, reproductive traits, and heat stress response traits in dairy cattle, as well as to identify key genetic markers available for selection. The results of this study could improve our understanding of genes with pleiotropic effects for complex traits and provide valuable molecular marker information for genetic selection in dairy cattle.

## 2. Results

### 2.1. Polymorphism Screening

In this study, a total of 42 variants were identified from polymorphism screening in 70 Holstein bulls, including 32 in *TRH*, 6 in *PRL*, and 4 in *PRLR* (Appendix A). Furthermore, the locus g.55885367T/C located in the 3′-flanking region of *TRH* is a novel variant, which has not been reported in Ensemble (https://asia.ensembl.org/, accessed on 1 January 2025). After the pilot analysis using 96 randomly selected samples, variants that did not meet the KASP primer design criteria were removed and only one variant was retained within each 200 bp range. Eventually, 13 variants within the genes *TRH* (8), *PRL* (3) and *PRLR* (2) were genotyped by KASP genotyping in 1152 Holstein cows. The allele frequencies and the corresponding *p*-value derived from a Hardy–Weinberg equilibrium test for 13 SNPs in 1152 cows are presented in Table 1. Except for g.35341821A/- located on *PRL*, and g.39077999G/A and g.39099173C/A located on *PRLR* (*p* < 0.05), all SNPs did not significantly deviate from Hardy–Weinberg equilibrium expectations. The distributions of SNP clustering (Figure 1) suggest that the genotyping qualities were generally good for all 13 SNPs.

In *TRH*, six SNPs (g.55891470A/G, g.55891369A/T, g.55891325G/A, g.55891270A/G, g.55891229T/A and g.55890228A/-) are located in the 5′-flanking region, one (g.55888602T/G) in the 5′-untranslated region (UTR), and one (g.55885455T/C) in the 3′-flanking region. For *PRL*, two SNPs (g.35341821A/- and g.35332944T/C) were located in the upstream regulatory region, one (g.35342351A/G) within an exon region. For *PRLR*, SNP g.39077999G/A and g.39099173C/A are located within an exon region and 3′-flanking region, respectively.

The LD between each pair of SNPs was estimated for the genes *TRH*, *PRL*, and *PRLR*, respectively. Two haplotype blocks were observed in *TRH* and *PRL*, and comprised five and two SNPs, respectively (Figure 2). The haplotype block of *TRH* consisted of SNP g.55891229T/A, g.55891270C/T, g.55891325T/C, g.55891369A/T and g.55891470C/T, including three haplotypes. The haplotype block constructed in *PRL* consisted of SNP g.35332944T/C and g.35342351A/G, including three haplotypes (Appendix A). In addition, there was a low LD between SNP g.39077999G/A and g.39099173C/A in *PRLR*, which were not considered as a haplotype block.

### 2.2. Association Analyses of SNP and Haplotype Blocks with Milk Performance

By running the linear model, the SNP and haplotype block-based association analyses reveal the genetic effects of *TRH*, *PRL*, and *PRLR* on milk performance traits, as shown in Figure 3 and Appendix A. Significant associations with milk performance traits (*p* < 0.05) were observed in all three candidate genes. Except for SNP g.55891270C/T and g.55891325T/C in *TRH*, all SNPs within *TRH*, *PRL*, and *PRLR* showed significant associations (*p* < 0.05) with at least one milk performance trait (fat yield, FY; milk yield, MY; protein yield, PY; and somatic cell score, SCS).

For the *TRH* gene, the SNP g.55888602A/C showed significant associations (*p* < 0.05) with all milk performance traits in the third lactation, including MY, FY, PY, and SCS. The cows with a C:C genotype in g.55888602A/C had a relatively high milk performance and low SCS compared to the other two genotypes. In addition, SNPs g.55890228A/-, g.55885455A/G and g.55891369A/T showed significant associations (*p* < 0.05) with eight (MY_2, MY_3, FY_1, FY_2, FY_3, PY_2, PY_3 and SCS_1), seven (MY_3, FY_1, FY_2, FY_3, PY_3, SCS_1 and SCS_3), and four (MY_2, FY_2, PY_2, and SCS_1) milk performance traits, respectively. The SNPs g.55891470C/T and g.55891229T/A had significant associations (*p* < 0.05) with the FY trait in the third lactation (Figure 3A).

Except for FY and SCS in the first lactation, the SNP g.35332944T/C within *PRL* shows significant associations (*p* < 0.05) with milk performance. In general, the C:C genotype in g.35332944T/C was a favorable genotype for milk performance traits with relatively higher milk performance potential and low SCS. The SNPs g.35341821A/- and g.35342351A/G had significant associations (*p* < 0.05) with FY, PY, and SCS. Both SNP g.39077999G/A and g.39099173C/A in *PRLR* had significant associations (*p* < 0.05) with SCS in all three lactations, and SNP g.39099173C/A also significantly associated (*p* < 0.05) with MY and FY in the third lactation.

The significant associations detected between the two haplotype blocks and 12 milk performance traits are also presented in Appendix A. Consistent with the results from SNP-based analyses, the haplotype block in *TRH* had significant associations with MY, FY, PY, and SCS. Similarly, the significant associations of the haplotype block in *PRL* also were found with FY, MY, PY, and SCS (Figure 3B).

### 2.3. Association Analyses of SNPs with Reproductive Traits

Regarding reproductive traits (Figure 3 and Appendix A), there are also significant associations between g.55888602A/C and various reproductive traits (age at the first calving in heifers, AFC; conception rate, CR; interval between the first and last insemination in heifers, IFL_H; interval between the first and last insemination in cows, IFL_C; calving ease in heifers, CE_H; calving ease in cows, CE_C; stillbirth in cows SB_C; and stillbirth in heifers, SB_H). However, g.55890228A/- was only found to be significantly associated with CE_C, IFL_H, and SB_H. In addition to g.55888602A/C, SNP g.55891470C/T, g.55891325T/C, g.55891270C/T, g.55891229T/A, and g.3907799G/A were also found to be significantly associated with AFC. Furthermore, 7 SNPs, 9 SNPs and 11 SNPs were found to be significantly associated with CE_H, CE_C, and SB_H, respectively. Among them, SNP g.55891470C/T, g.55891325T/C, g.55891270C/T, g.55891229T/A, g.55888602A/C, and g.55885455A/G are all associated with the above three reproductive traits. However, none of the 13 SNPs were associated with age at the first service in heifers (AFS) (Figure 3A).

The significant associations of haplotype blocks detected with nine reproductive traits are presented in Figure 3B and Appendix A. Consistent with the results from SNP-based analyses, haplotype block in *TRH* had significant associations with AFC, CE_H, CE_C, and SB_H in reproductive traits, and significant associations of a haplotype block in *PRL* were found with CR in reproductive traits.

### 2.4. Association Analyses of SNPs with Heat Stress Response Traits

As presented in Figure 3 and Appendix A, the SNP-level association analyses showed that all SNPs of genes *TRH*, *PRL*, and *PRLR* exhibited significant associations (*p* < 0.05) with a drooling score (DS), except for g.35342351A/G in *PRL*. Furthermore, the SNPs g.55888602A/C and g.55885455A/G within *TRH*, and SNP g.35342351A/G within *PRL* had significant associations (*p* < 0.05) with rectal temperature (RT). The SNPs g.35341821A/- and g.35342351A/G within *PRL* and SNP g.39099173C/A within *PRLR* had significant associations (*p* < 0.05) with respiratory rate (RR) (Figure 3A).

The haplotype-based association analyses found that the haplotype block in *TRH* was significantly associated with DS, and the haplotype block in *PRL* was significantly associated with three heat stress response traits (Figure 3B and Appendix A).

### 2.5. TRH, PRL, and PRLR Expressed in Different Tissues and Associated with Multiple Traits

To understand the expression differences of *TRH*, *PRL*, and *PRLR* in different tissues in cattle, full tissue expression profiles were constructed using gene expression data from each tissue in the cattleGTEx database (https://cgtex.roslin.ed.ac.uk, accessed on 7 June 2024). The result showed that each of these genes is expressed specifically in different tissues. *TRH* is highly expressed in the hypothalamus and frontal cortex (Figure 4A). *PRL* is highly expressed in the hypothalamus, pituitary gland, and testis (Figure 4B). *PRLR* is highly expressed in the liver and in the mammary gland (Figure 4C).

To further explore the roles of *TRH*, *PRL*, and *PRLR* in cattle complex traits, QTLs were mapped in the Animal QTL database (www.animalgenome.org/cgi-bin/QTLdb/index, accessed on 13 June 2024) (Appendix A). As shown in Figure 5, 143 QTLs were identified in these three genes, including 11 in *TRH*, 15 in *PRL*, and 117 in *PRLR*. In addition to milk performance traits, the QTLs in these genes have an association with reproductive traits. In addition, g.39099173C/A of *PRLR* was found to be a significant eQTL in the cattleGTEx database (https://cgtex.roslin.ed.ac.uk, accessed on 7 June 2024) (Appendix A). In RNA-Seq results indicated g.39099173C/A as an eQTL affecting *PRLR* mRNA expression levels; specifically, the AA allele showed dominant expression (Appendix A).

## 3. Discussion

In this study, we investigated the genetic effects of *TRH*, *PRL* and *PRLR* on milk performance, reproductive, and heat stress response traits in cattle using population-level association analyses. Among 13 SNPs of genes *TRH*, *PRL*, and *PRLR* genotyped in the Holstein population, 12 SNPs were significantly associated with different reproductive traits, with the greatest number of significant SNPs in gene *TRH* for CE_C (8 SNPs) and SB_H (8 SNPs). Meanwhile, there was a large number of SNPs that were significantly associated with AFC (4 SNPs) and CE_H (6 SNPs) in *TRH*.

TRH is a neuropeptide secreted by the hypothalamus that stimulates the secretion of thyroid-stimulating hormone by pituitary cells and stimulates the release of prolactin in a dose-dependent manner. In mammals, TRH acts as a prolactin-releasing factor, capable of regulating prolactin synthesis and secretion via a complex cascade of diverse signaling pathways. In this study, we also found that *TRH* is specifically highly expressed in the hypothalamus and frontal cortex of Holstein cows. Meanwhile, as a downstream regulated factor, *PRL* also has a significantly high expression in the hypothalamus and pituitary. It has been reported that heat stress activates the hypothalamic–pituitary–adrenal (HPA) axis. The hypothalamus secretes corticotropin-releasing hormone (CRH), which stimulates the pituitary gland to secrete adrenocorticotropic hormone (ACTH). In turn, ACTH stimulates the adrenal cortex to synthesize and secrete cortisol. This process adversely affects the hypothalamic–pituitary–gonadal axis and the female estrous cycle [24]. Our study provides a new genetic insight into how cows respond to heat stress and the latter affects reproductive and milk performance traits. *PRLR* was significantly highly expressed in the mammary gland, which seems to provide some explanation for the regulatory relationship between *TRH*–*PRL*–*PRLR* at the tissue level. The hypothalamus stimulates the pituitary gland to secrete PRL through the secretion of TRH, and PRL exerts its biological effects by binding to PRLR on the surface of breast cells.

A previous study in rats demonstrated that *TRH* modulates prolactin synthesis and secretion through microRNA (specifically, miR-126a-5p) [25]. Additionally, feeding TRH to sows increases serum thyroxine levels, elevates growth hormone and prolactin concentrations, and markedly enhances milk performance and weaning weight on day 20 postpartum. However, this treatment is also associated with a delayed onset of estrus post-weaning [26]. In addition, Lin et al. (1989) suggest that cold stress can increase the TRH level in the hypothalamus of rats [27]. This issue is consistent with our findings that all SNPs of the *TRH* gene are significantly genetically associated with the three traits (milk performance, reproductive, and heat stress response traits). Especially, the SNP (g.55888602A/C) displays a significant association with all milk performance traits and nearly all reproductive and heat stress response traits. Additionally, g.55885455A/G and g.55890228A/- are also significantly associated with the majority of these traits, providing further insight into the genetic mechanisms and underlining the crucial role of *TRH* in reproduction, milk performance, and the adaptive response to both cold and heat stress.

In this study, three SNPs of *PRL* gene, including g.35341821A/-, g.35342351A/G, and g.35332944T/C, were found in the Holstein population. Through genotyping and association analysis, all three SNPs, except for g.35342351A/G, showed significant associations with several reproductive traits. PRL has been shown to play an important role in regulating mammary gland development and lactation, regulating gonadal development, regulating immune function, participating in stress response, and regulating the proliferation and division of somatic cells in mice [28,29]. There are also many discoveries in the study of *PRL* gene polymorphisms, but only some of its many SNP have been proven to have a significant genetic association with milk performance traits in Bufala Mediterranea Italiana [21]. The study for reproductive and heat stress response traits is mainly focused on its functional role as a pituitary hormone [13,30]. In this study, the association analysis between gene polymorphisms in *PRL* and milk performance, and reproductive traits and heat stress response demonstrated its important genetic effects on these traits, and the SNPs screened by this study can be used as genetic markers for genetic selection of reproductive traits and heat stress response in high-yielding dairy cows.

In Finnish Ayrshire dairy cattle, *PRLR* S18N polymorphism was found to be significantly associated with protein and fat yield [31]. In addition, polymorphisms in the *PRLR* gene were found to be significantly associated with growth traits in Nanyang breeds [32]. In Egyptian buffaloes, *PRLR* gene polymorphisms have been demonstrated to be genetically associated with productive and reproductive traits, and milk performance traits [23,33]. In Italian Mediterranean river buffaloes, Cosenza et al. (2018) screened multiple polymorphisms of the *PRLR* gene, and proposed that the *PRLR* gene is a good candidate gene for the genetic selection of milk performance-related traits in buffalo [34]. Both the above studies and this study show that the *PRLR* gene plays an important role in milk performance and reproductive traits in cattle, but the genetic role of *PRLR* in the cold and heat stress response of dairy cows has not been reported. This study found significant associations between the SNPs g.39099173C/A and g.39077999G/A within *PRLR* and heat stress response traits, suggesting that the *PRLR* gene may play a crucial role in assisting cows to adapt to and cope with the challenging environmental conditions associated with heat stress.

## 4. Materials and Methods

### 4.1. SNP Identification in 70 Bulls

The DNA was extracted from frozen semen samples of 70 Holstein bulls (Beijing Dairy Cattle Center, Beijing, China) using a standard phenol-chloroform method, and the DNA concentration was then diluted to 200 ng/µL with TB buffer.

Using Premier 5.0 and Oligo 7.0 (Sangon Biotech Co., Ltd. Shanghai, China), a total of 28 pairs of primers were designed for amplification of all exons and flanking regions (within 2 kb distance of 5′ and 3′ end) in three candidate genes, including 6 for *PRL*, 13 for *PRLR* and 9 for *TRH* (Appendix A). These primers were then synthesized at Sangon Biotech (Shanghai, China). We randomly mixed DNA samples (200 ng/µL) into three pools (one pool contained samples from 30 bulls, and two pools contained 20 samples each) and then used pooled samples for the subsequent polymerase chain reaction (PCR) amplification (ABI3730XL DNA analyzer (Applied Biosystems, Foster, CA, USA)). Sequencing data were aligned to the reference genome (UMD 3.1) using Ensemble (https://asia.ensembl.org/, accessed on 16 December 2020) and examined for potential polymorphism using Chromas (http://technelysium.com.au/wp/chromas/, accessed on 18 May 2021) and DNAman (https://www.lynnon.com/dnaman.html, accessed on 3 June 2021).

### 4.2. Genotyping and Phenotyping in 1152 Cows

We considered 1152 Holstein cows from eight dairy farms (Beijing Sunlon Livestock Development Co., Ltd., Beijing, China), and the blood samples and phenotypic data of all these cows were collected. Then, DNA samples of 1152 Chinese Holstein cows were extracted from the whole blood samples using a TIANamp Blood DNA Kit (TIANgen, Beijing, China). To shortlist variants included for KASP (Kompetitive allele-specific PCR) genotyping, a pilot analysis was performed in 96 (out of 1152) randomly selected samples. Then, all 1152 cows were genotyped using the KASP method. A chi-square test was used to determine whether allelic frequencies of any variant deviated from the Hardy–Weinberg equilibrium. Bonferroni correction was applied to control for familywise false positives.

In this study, four milk performance traits, nine reproductive traits and three heat stress response traits were analyzed, covering the main traits of cow milk performance and reproductive performance and physiological indicators of Holstein cows under heat stress [35,36]. The milk performance traits included the milk yield (MY), fat yield (FY), protein yield (PY), and somatic cells score (SCS), with each trait including data from the first (MY_1, FY_1, PY_1, and SCS_1), second (MY_2, FY_2, PY_2 and SCS_2), and third (MY_3, FY_3, PY_3 and SCS_3) lactations. The genetic evaluation for these traits was based on test-day data. Reproductive traits included age at the first service (AFS), calving (AFC), conception rate (CR) at first insemination, the interval from the first to last insemination in heifers (IFL_H) and cows (IFL_C), stillbirth in heifers (SB_H) and cows (SB_C), and calving ease in heifers (CE_H) and cows (CE_C); and heat stress response traits included the rectal temperature (RT), respiratory rate (RR), and drooling score (DS). Estimated breeding values (EBVs) for each trait, derived from single-trait animal models by routine genetic evaluation (Independent Innovation League of Dairy Breeding, Beijing, China), were used as response variables for association analyses in the current study [37,38,39,40]. The details of routine genetic evaluation for these traits are provided in Appendix A. Descriptive statistics and distribution of EBVs for the above 16 traits are presented in Appendix A, respectively.

### 4.3. Construct Haplotypes Based on LD Structures

For each candidate gene, haplotype blocks were constructed based on the LD structures of identified SNPs using Haploview4.0 software [41]. Haplotypes with relative frequency less than 5% were merged into one group (Figure 2).

### 4.4. Association Analysis

Association analyses of each identified SNP or haplotype block with milk performance, reproductive, and heat stress response traits were performed using the general linear model (GLM) procedure of SAS 9.2 (SAS Institute, Cary, NC, USA). For both SNP- and haplotype block-based analyses, Bonferroni correction was used to control for false positives resulting from multiple testing (*p* < 0.05). The model used for the association analysis of each trait is as follows:y=μ+G+e
where y is the individual EBV; µ is the overall mean; G is the fixed effect of the genotype or haplotype block; and e is the random residual effect. The proportion of the phenotypic variance ($\sigma_{P}^{2}$) explained by the each identified SNP was calculated as ($\frac{\sigma_{g}^{2}}{\;\sigma_{P}^{2}}$), where $\sigma_{g}^{2}$ was the variance explained by SNP and $\sigma_{P}^{2}$ was the phenotypic variance [42].

### 4.5. Functional Annotation Analysis and Visualization

The quantitative trait locus (QTL) information covering all traits for cattle (ARS UCD1.2) was downloaded from the Animal QTL database Release 55 (https://www.animalgenome.org/cgi-bin/QTLdb/index, accessed on 13 June 2024), and the mapping analyses of gene *TRH*, *PRL* and *PRLR* were performed using the cattle QTL database. Whole-body tissue expression information and expression quantitative trait locus (eQTL) information for these three genes in this study were obtained from the cattleGTEx database (https://cgtex.roslin.ed.ac.uk, accessed on 7 June 2024). Results visualization was performed using the R package ggplot2. The quality of the raw reads from the RNA-seq data was assessed using FastQC v0.11.9 (https://www.bioinformatics.babraham.ac.uk/projects/fastqc/, accessed on 13 September 2023). Subsequently, an NGS QC Toolkit v2.3.3 was used to remove adaptors and filter out poor-quality reads [43]. The clean reads were aligned to the Bos taurus reference genome (version: ARS-UCD1.2.109) using the Hisat2 v2.2.1 [44]. The aligned SAM files were then converted into BAM files using SAMtools v1.9 (https://github.com/samtools/samtools/releases/, accessed on 16 September 2023), followed by quantitative analysis with Featurecounts. Then, the TPM (transcripts per million) were used to measure gene or transcript expression levels. And complementary statistical evaluations were conducted using GraphPad Prism version 9.3.1.

## 5. Conclusions

In this study, we screened the genetic polymorphisms of three important hormone-related genes, *TRH*, *PRL*, and *PRLR*, and investigated their genetic effects on milk performance, reproductive, and heat stress response traits in dairy cows. Of the 13 SNPs screened in Holstein cows, all SNPs were significantly associated with at least one of the three important traits. The SNP g.55888602A/C and g.55885455A/G of the *TRH* gene, which were significantly associated with multiple milk performance, reproductive, and heat stress response traits, are key molecular markers for genetic selection of reproductive traits and heat stress response in high-yielding dairy cows. The QTL enrichment analysis and the tissue expression profiles analysis also demonstrated the important roles of the three genes in milk performance and reproductive traits in dairy cows. By integrating key molecular markers found by the current study into genomic prediction, the prediction reliability of breeding values can be improved, and therefore, accelerate the genetic improvement program of high-yield dairy cattle.

## Figures and Tables

**Figure 1 ijms-26-01963-f001:**
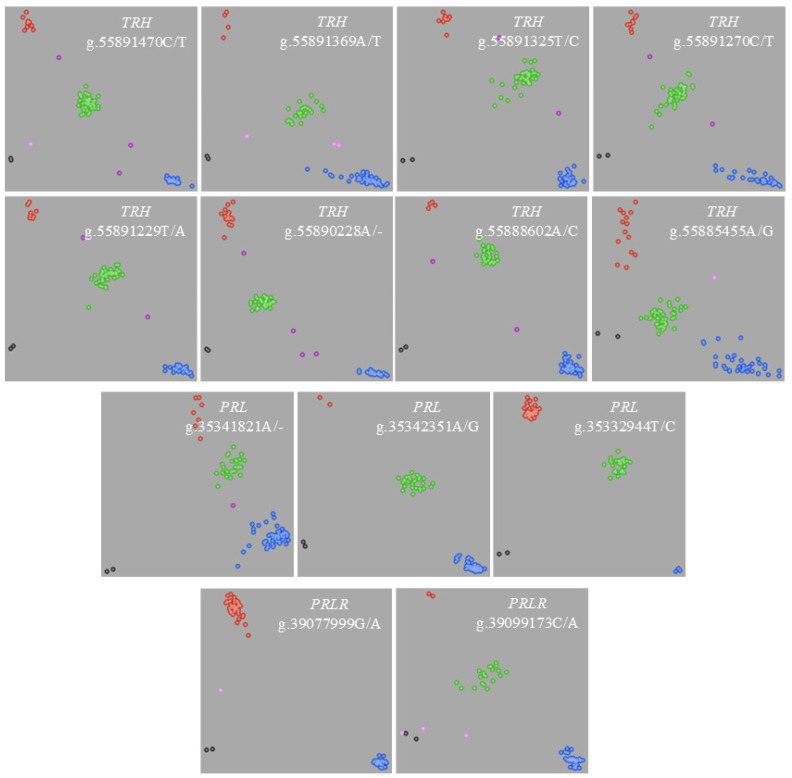
Distributions of SNP clustering for the identified 13 SNPs within *TRH*, *PRL*, and *PRLR* using KASP genotyping method. The red and blue represent homozygous genotype; the green represents heterozygous genotype; the pink represents no or a weak signal; the purple represents signal for no genotyping; and the black represents blank control.

**Figure 2 ijms-26-01963-f002:**
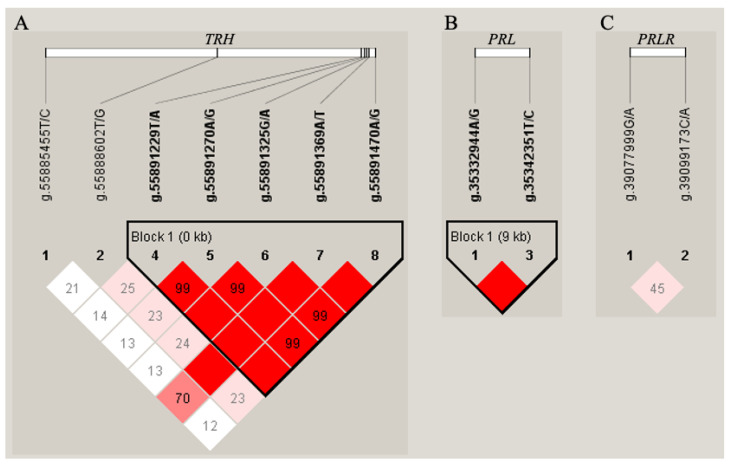
Haplotype blocks constructed based on linkage disequilibrium (LD) for gene *TRH* (**A**), *PRL* (**B**), and *PRLR* (**C**) in 1152 Holstein cows.

**Figure 3 ijms-26-01963-f003:**
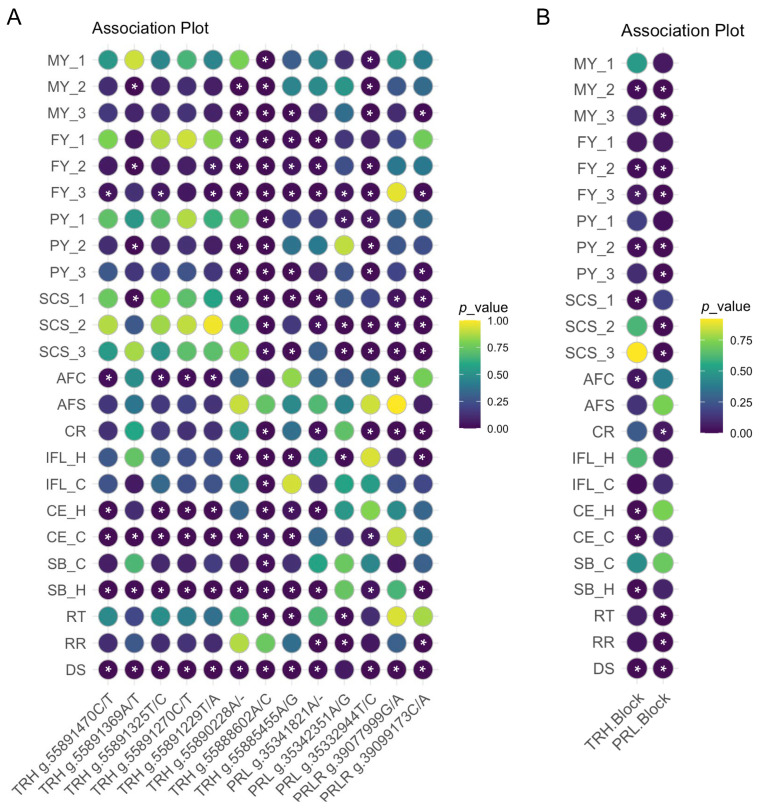
Associations of SNPs and haplotype blocks within gene *TRH*, *PRL*, and *PRLR* with milk performance, reproduction, and heat stress response traits in Holstein cows. (**A**) Associations of SNPs within gene *TRH*, *PRL*, and *PRLR*. (**B**) Association of haplotype blocks within gene *TRH*, *PRL*, and *PRLR*. Milk performance traits included milk yield (MY), fat yield (FY), protein yield (PY), and somatic cells score (SCS), with each trait including data from the first (MY_1, FY_1, PY_1 and SCS_1), second (MY_2, FY_2, PY_2 and SCS_2) , and third (MY_3, FY_3, PY_3 and SCS_3) lactation. Reproductive traits included age at the first service (AFS) and calving (AFC), the interval from the first to last insemination in heifers (IFL_H) and cows (IFL_C), conception rate of first insemination (CR), calving ease in heifers (CE_H) and cows (CE_C), and stillbirth in heifers (SB_H) and cows (SB_C). Heat stress response traits included rectal temperature (RT), respiratory rate (RR), and Drooling score (DS). The “*” indicates a *p*-value less than 0.05.

**Figure 4 ijms-26-01963-f004:**
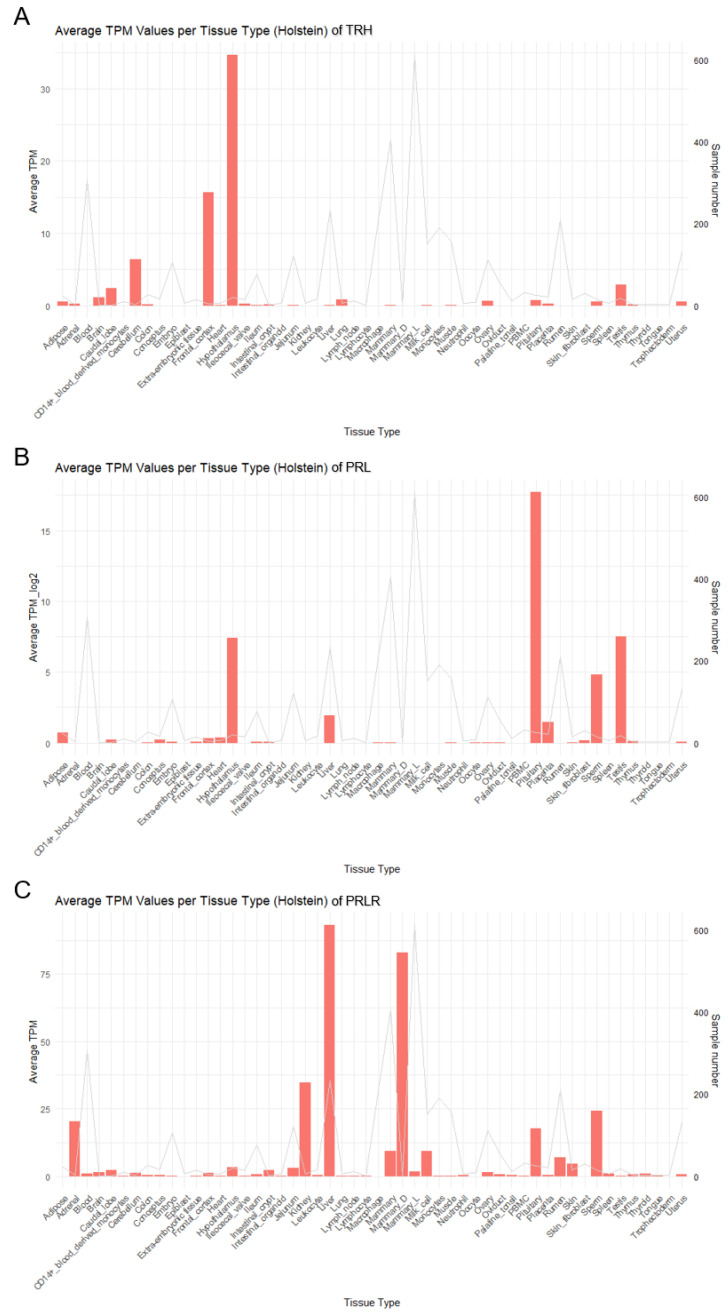
The whole-body tissue expression profiles of *TRH* (**A**), *PRL* (**B**), and *PRLR* (**C**) in Holstein cattle. The bars in the combined graph represent transcripts per million (TPM) values, and the line represents the sample size.

**Figure 5 ijms-26-01963-f005:**
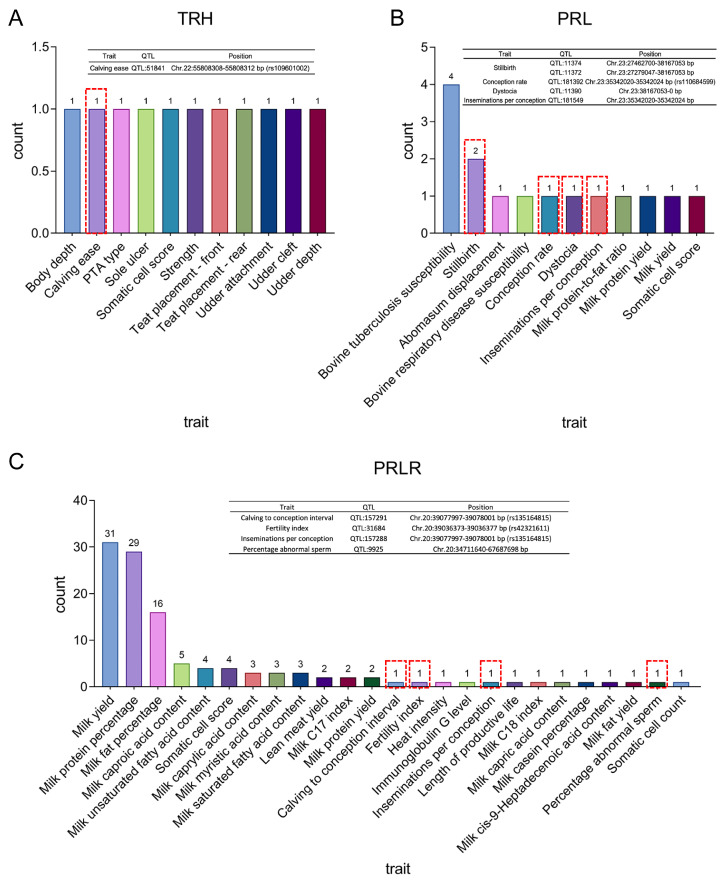
The multiple traits-associated quantitative trait loci (QTLs) for gene *TRH* (**A**), *PRL* (**B**), and *PRLR* (**C**) in Holstein cattle. The tables in the figure show the information about the loci related to reproductive traits in the red box.

**Table 1 ijms-26-01963-t001:** Detailed information on 13 SNPs within the gene *TRH*, *PRL*, and *PRLR* in 1152 Holstein cows.

Gene	SNP	Genotype	Number of Individuals	Genotypic Frequency	Allele	Allelic Frequency	*p* Value, (df = 2) ^1,2^
*TRH*	g.55891470C/T	C:C	174	0.154	C	0.383	0.615
		C:T	519	0.459	T	0.617	
		T:T	438	0.387			
*TRH*	g.55891369A/T	T:T	785	0.703	T	0.843	0.161
		A:T	312	0.280	A	0.157	
		A:A	19	0.017			
*TRH*	g.55891325T/C	T:T	172	0.153	T	0.381	0.576
		T:C	515	0.457	C	0.619	
		C:C	440	0.390			
*TRH*	g.55891270C/T	C:C	172	0.153	C	0.382	0.664
		C:T	518	0.460	T	0.618	
		T:T	437	0.388			
*TRH*	g.55891229T/A	T:T	170	0.151	T	0.381	0.720
		T:A	518	0.460	A	0.619	
		A:A	437	0.388			
*TRH*	g.55890228A/-	A:A	118	0.108	A	0.327	0.972
		A:-	476	0.437	-	0.673	
		-:-	496	0.455			
*TRH*	g.55888602A/C	A:A	753	0.662	A	0.811	0.680
		C:A	339	0.298	C	0.189	
		C:C	45	0.040			
*TRH*	g.55885455A/G	A:A	424	0.378	A	0.621	0.540
		G:A	545	0.486	G	0.379	
		G:G	152	0.136			
*PRL*	g.35341821A/-	A:A	103	0.092	A	0.255	0.000
		A:-	367	0.327	-	0.745	
		-:-	653	0.581			
*PRL*	g.35342351A/G	A:A	900	0.790	A	0.888	0.739
		G:A	222	0.195	G	0.112	
		G:G	17	0.015			
*PRL*	g.35332944T/C	C:C	673	0.591	C	0.772	0.760
		C:T	410	0.360	T	0.228	
		T:T	55	0.048			
*PRLR*	g.39077999G/A	A:A	725	0.635	A	0.644	0.000
		A:G	22	0.019	G	0.356	
		G:G	395	0.346			
*PRLR*	g.39099173C/A	A:A	35	0.031	A	0.145	0.036
		A:C	262	0.230	C	0.855	
		C:C	844	0.740			

^1^ Hardy–Weinberg equilibrium test; ^2^ Chi-square value, χ^2^.

## Data Availability

The data supporting the findings of this study are available in the Appendix A of this article. Additional questions can be addressed to the corresponding author.

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
