# Peer review of "Association of Genes TRH, PRL and PRLR with Milk Performance, Reproductive Traits and Heat Stress Response in Dairy Cattle"

_ijms, 2025, doi:10.3390/ijms26051963_

Round 1
Reviewer 1 Report
Comments and Suggestions for Authors
The work is valuable, but has some shortcomings, such as Figure 2, which lacks a description. There is a doubt about the use of males and not females in the study (considering the physiological part and not just the genetic/heritage part).
It would be worthwhile to be able to explain the effect that these genes have on the physiological regulation of the tissues where they were found to be most closely related to better understand how they would regulate responses on milk yield, reproduction and heat stress response traits

Author Response
Comments 1: The work is valuable, but has some shortcomings, such as Figure 2, which lacks a description. There is a doubt about the use of males and not females in the study (considering the physiological part and not just the genetic/heritage part).
Response: Thank you for pointing out the value of this study. Firstly, regarding the issue of lacking descriptions for the Figures, we have made modifications to both Figure 2 and Figure 1 by adding descriptions of the corresponding genes to enhance their readability. Secondly, due to the widespread application of artificial insemination technology in dairy industry, a bull can have many offspring, so the genetic diversity and representation of breeding bulls is much higher than that of cows. By collecting DNA from a group of bulls, it is possible to quickly screen the population for potential polymorphisms in target genes, and then to perform batch genotyping of found polymorphisms in the large female population, which is more efficient in terms of cost and technology.
Comments 2: It would be worthwhile to be able to explain the effect that these genes have on the physiological regulation of the tissues where they were found to be most closely related to better understand how they would regulate responses on milk yield, reproduction and heat stress response traits.
Response: The characteristically high expression of these genes in different tissues and their physiological regulatory roles have been explained accordingly in our discussions (Line236 - Line 249). Thanks for your comments.
Comments 3: The "highlighting" and "underlining" marked in the manuscript.
Response: I have carefully reviewed the feedback you provided and have endeavored to address the issues in the suggestions. However, I encountered some difficulty in fully understanding the exact meaning of the "highlighting" and "underlining" marked in the manuscript. Nonetheless, I have repeatedly reviewed and partially revised the labeled sections in an effort to meet the reviewers' high standards.

Reviewer 2 Report
Comments and Suggestions for Authors
The authors identified some SNPs in three genes that were associated with milk production and reproductive traits, as well as heat stress response in the Chinese Holstein population. This is an interesting work. Nonetheless, the following issues need to be further addressed:
1. Line 16 or 17: these three genes or the three genes? please fix it.
2.Line37: why you add the word "respectively"? Aren't both these traits important economic and functional traits?
3. milk performance traits or milk production traits? please fix them in the manuscript
4. Line 89: Supplementary Table S3?why not is the Supplementary Table S1? Authors should cite the attachments in order
5.Line 125: In the Figure 2, the author should add the subtitle for each picture. the gene name is more suitable for subtitle
6. Line 130 or 153: Table S4 or Supplementary Table S4? please fix it
7.Line 169-181: the author should cite the Figure or Table name. these results reference which Figure or Table?
8. Line 201: please provide the full name for cGTEx?How about the URL for this database?
9.Line 212: I have no idea which public databases were used in this study?
10. ng/µl or ng//µL? please fix them
11. Line 327: please add the information on the threshold level for association analysis
12. Line 341: please provide the database version
13. Line 348-355: please cited the reference for these softwares
Comments on the Quality of English LanguageNot applicable
Author Response
Comments 1: Line 16 or 17: these three genes or the three genes? please fix it.
Response: “these three genes” has been modified to “the three genes”.
Comments 2: Line37: why you add the word "respectively"? Aren't both these traits important economic and functional traits?
Response: Indeed, both of these traits are important economic and functional traits, so we removed the word "respectively".
Comments 3: milk performance traits or milk production traits? please fix them in the manuscript.
Response: We have modified the full text “milk production traits” to “milk performance traits”.
Comments 4: Line 89: Supplementary Table S3?why not is the Supplementary Table S1? Authors should cite the attachments in order.
Response: We have modified ”Supplementary Table S3” to “Supplementary Table S1”, and adjusted the Supplementary table cite for the full text.
Comments 5: Line 125: In the Figure 2, the author should add the subtitle for each picture. The gene name is more suitable for subtitle.
Response: We have modified both Figure 2 and Figure 1 to improve their readability by adding descriptions of the corresponding genes. Thanks for your comments.
Comments 6: Line 130 or 153: Table S4 or Supplementary Table S4? please fix it.
Response: “Table S4” has been modified to “Supplementary Table S4”.
Comments 7: Line 169-181: the author should cite the Figure or Table name. these results reference which Figure or Table?
Response: These results cite Figure3 and Supplementary Table S3, and we have modified this section to add the appropriate citations.
Comments 8: Line 201: please provide the full name for cGTEx?How about the URL for this database?
Response: The “cGTEx” in the full text has been modified to the full name “cattleGTEx”. At the same time, we added the URL of cattleGTEx database everywhere it appeared.
Comments 9: Line 212: I have no idea which public databases were used in this study?
Response: We provide further clarification on the databases from which the relevant results were derived.
Comments 10: ng/µl or ng//µL? please fix them.
Response: “ng/µl” has been modified to “ng/µL”.
Comments 11: Line 327: please add the information on the threshold level for association analysis.
Response: The threshold for association analysis is 0.05. Specific to different genotypes/haplotypes, multiple testing were performed. We have added it to the corresponding position in”4.4. Association analysis”. (Line 348)
Comments 12: Line 341: please provide the database version.
Response: The version of the Animal QTL database is Release 55 and we have added the information in the corresponding position.
Comments 13: Line 348-355: please cited the reference for these softwares.
Response: For the software used, we have added relevant reference citations.

Reviewer 3 Report
Comments and Suggestions for Authors
1. The locus g.55885367T/C, located in the 3'-flanking region of TRH, is a novel variant that is only revealed in lane 20 and lane 90 of this study. However, this variant shows no significant correlation with the milk performance, reproductive traits, or heat stress response traits analyzed. It is suggested that this variant should not be included in the Abstract, as it does not contribute to the main focus of the study.
2. In Lane 97-99, the manuscript states, "Except for g.39077999G/A located on PRLR (P < 0.05), all SNPs did not significantly deviate from Hardy-Weinberg equilibrium expectations." However, Table 1 shows that the p-values for TRH (g.55891369A/T), PRL (g.35341821A/-), and PRLR (g.39099173C/A) are all less than 0.05. This discrepancy should be addressed, and the text should be revised to reflect the correct information, as it is inconsistent with the data presented in Table 1.
3. In Lane 108, there is a discrepancy regarding the number of Holstein cows used in the analysis. Table 1 shows results based on either 1150 or 1152 Holstein cows. Please clarify and confirm the correct number of cows used in the analysis.
4. In Lane 120-121, the manuscript states, "The haplotype block constructed in PRL consisted of SNP g.35332944C/T and g.35342351A/G, including six haplotypes." However, it is unclear whether there are three or six haplotypes. Please verify and clarify the correct number of haplotypes for this haplotype block.
5. In Lane 124, it is recommended to label TRH, PRL, and PRLR separately in Figure 2. This will help readers easily distinguish between the different genes and their corresponding data.
6. In Lane 361-364, the Conclusion states, "The SNP g.55888602A/C of the TRH gene and g.35332944C/T of the PRL gene, which were significantly associated with multiple milk performance, reproductive, and heat stress response traits, are key molecular markers for genetic selection of reproductive traits and heat stress response in high-yielding dairy cows." However, the main text does not report any significant associations of g.35332944C/T of the PRL gene with reproductive and heat stress response traits. Additionally, the Abstract mentions only the important roles of SNP g.55888602A/C in TRH for all traits, providing important molecular markers for genetic selection of high-yielding dairy cows. This inconsistency between the Abstract and the Conclusion needs to be addressed and clarified. Please revise to ensure consistency throughout the manuscript.
Author Response
Comments 1: The locus g.55885367T/C, located in the 3'-flanking region of TRH, is a novel variant that is only revealed in lane 20 and lane 90 of this study. However, this variant shows no significant correlation with the milk performance, reproductive traits, or heat stress response traits analyzed. It is suggested that this variant should not be included in the Abstract, as it does not contribute to the main focus of the study.
Response: Thanks for your comments. We have removed this variant introduction from the Abstract.
Comments 2: In Lane 97-99, the manuscript states, "Except for g.39077999G/A located on PRLR (P < 0.05), all SNPs did not significantly deviate from Hardy-Weinberg equilibrium expectations." However, Table 1 shows that the p-values for TRH (g.55891369A/T), PRL (g.35341821A/-), and PRLR (g.39099173C/A) are all less than 0.05. This discrepancy should be addressed, and the text should be revised to reflect the correct information, as it is inconsistent with the data presented in Table 1.
Response: Indeed, we rechecked the relevant data, updated the p-values, and made some changes to this section. “Except for g.35341821A/- located on PRL, and g.39077999G/A and g.39099173C/A located on PRLR (P < 0.05), all SNPs did not significantly deviate from Hardy-Weinberg equilibrium expectations.” (Line 94 - Line 96)
Comments 3: In Lane 108, there is a discrepancy regarding the number of Holstein cows used in the analysis. Table 1 shows results based on either 1150 or 1152 Holstein cows. Please clarify and confirm the correct number of cows used in the analysis.
Response: Indeed, the number of Holstein cows used for the analysis was based on 1,152, and we have made a correction.
Comments 4: In Lane 120-121, the manuscript states, "The haplotype block constructed in PRL consisted of SNP g.35332944C/T and g.35342351A/G, including six haplotypes." However, it is unclear whether there are three or six haplotypes. Please verify and clarify the correct number of haplotypes for this haplotype block.
Response: Thanks for your comments. There was a clerical error here, it should be SNP g.35332944T/C. However, it is true that the haplotype block constructed in PRL including three haplotypes. The SNP g.35342351A/G has three genotypes, and SNP g.35332944T/C has three genotypes, and they existed in a total of three haplotypes CA, TA, and CG, and six combinations (haplotype combination), AACC (485), AACT (357), AATT (55), GACC (170), GACT (56), and GGCC (17) in the population of 1152 Holstein cows in this study.
Comments 5: In Lane 124, it is recommended to label TRH, PRL, and PRLR separately in Figure 2. This will help readers easily distinguish between the different genes and their corresponding data.
Response: We have modified both Figure 2 and Figure 1 to improve their readability by adding label of the corresponding genes separately. Thank you.
Comments 6: In Lane 361-364, the Conclusion states, "The SNP g.55888602A/C of the TRH gene and g.35332944C/T of the PRL gene, which were significantly associated with multiple milk performance, reproductive, and heat stress response traits, are key molecular markers for genetic selection of reproductive traits and heat stress response in high-yielding dairy cows." However, the main text does not report any significant associations of g.35332944C/T of the PRL gene with reproductive and heat stress response traits. Additionally, the Abstract mentions only the important roles of SNP g.55888602A/C in TRH for all traits, providing important molecular markers for genetic selection of high-yielding dairy cows. This inconsistency between the Abstract and the Conclusion needs to be addressed and clarified. Please revise to ensure consistency throughout the manuscript.
Response: The SNP g.35332944C/T in the PRL gene was initially a clerical error, and we have corrected it to g.35332944T/C. For consistency between the abstract and the conclusions, we have removed mention of it from the abstract, even though this SNP has significant associations with reproduction and heat stress traits.

Round 2
Reviewer 2 Report
Comments and Suggestions for Authors
no comments
Comments on the Quality of English Languagenot applicable
Reviewer 3 Report
Comments and Suggestions for Authors
The manuscript has been significantly improved.